# Extending the range of graph neural networks with global encodings

Alessandro Caruso [1,6], Jacopo Venturin [1,2,6], Lorenzo Giambagli [1], Edoardo Rolando [1], Zakariya El-Machachi[1], Frank Noé [1,2,3,4] ✉ & Cecilia Clementi [1,4,5] ✉

Graph Neural Networks (GNNs) are routinely used in molecular physics, social sciences, and economics to model many-body interactions in graph-like systems. However, GNNs are inherently local and can suffer from information flow bottlenecks. This is particularly problematic when modeling large molecular systems, where dispersion forces and local electric field variations drive collective structural changes. We introduce RANGE, a model-agnostic framework that employs an attention-based aggregation-broadcast mechanism that significantly reduces oversquashing effects, and achieves remarkable accuracy in capturing long-range interactions with linear scaling. Notably, RANGE integrates attention with positional encodings and regularization to dynamically expand virtual representations in virtual-node message-passing implementations. Across multiple state-of-the-art baselines, RANGE consistently restores long-range information, enabling the models to correctly predict electrostatic and dispersion-driven behavior even in out-of-distribution extrapolation tasks, where other unmodified baselines fail. Compared with other long-range paradigms, RANGE achieves the highest accuracy while requiring significantly less computational overhead, and it enables stable and scalable molecular dynamic simulations. RANGE offers accurate and efficient modeling of long-range interactions for simulating large molecular systems.

In the last decade, Message Passing Neural Networks (MPNNs) and, more generally, Graph Neural Networks (GNNs) have been established as a powerful and flexible approach to learning from graph-structured data[1–3]. In GNNs, the graph nodes act as artificial neurons, and local many-body information is aggregated at each message-passing step by updating node weights with messages received from direct neighbors. By repeating such message-passing steps multiple times, the field of view of each node expands to higher-order neighbors. In molecular science, GNNs have been found particularly useful in the development of Machine-Learned Force-Fields (MLFFs), where the nodes correspond to particles with a physical location in three-dimensional space - either corresponding to atoms in an atomistic force-field[4–10], or beads in a coarse-grained (CG) force-field[11–14]. These MLFFs are trained using energies or forces of molecules and configurations derived from a trusted ground truth, such as quantum-chemistry calculations or classical all-atom simulations. MLFFs have evolved in recent years, reflecting new trends and the fast development of network architectures in machine learning. Examples include the incorporation of physical symmetries and equivariances[4,8], attention mechanisms[10,15,16], and the integration of physics-based functional forms[7,17]. The main limitation of GNN-based MLFFs is their inherent locality. The neighborhood of

[1]Department of Physics, Freie Universität Berlin, Arnimallee 12, Berlin, Germany. [2]Department of Mathematics and Computer Science, Freie Universität Berlin, Arnimallee 12, Berlin, Germany. [3]AI4Science, Microsoft Research, Karl-Liebknecht Str. 32, Berlin, Germany. [4]Department of Chemistry, Rice University, 6100 Main Street, Houston, TX, USA. [5]Center for Theoretical Biological Physics, Rice University, 6100 Main Street, Houston 77005 TX, USA. [6]These authors contributed equally: Alessandro Caruso, Jacopo Venturin. ✉e-mail: frank.noe@fu-berlin.de; cecilia.clementi@fu-berlin.de

each particle node is usually defined to be all the other particles within a cutoff radius. In each message-passing step, information is exchanged within this radius. The field of view of each graph node is thus limited by the cutoff radius multiplied by the number of message-passing steps. While most MLFFs use cutoff radii of a few Ångströms to limit the computational cost of the message-passing operations, long-ranged electrostatic interactions can span several tens of Ångströms, in particular at interfaces such as biomembranes or in low-dielectric solvents[18,19]. The brute-force approach of extending the number of message-passing steps leads to highly correlated node representations, averaging out the information that, as it travels across the network, is further deteriorated by the presence of topological bottlenecks[20]. These two well-known limitations of GNNs, with many message-passing steps and large cutoffs, respectively known as oversmoothing and oversquashing, significantly impair long-range message-passing. Moreover, extending the cutoff radius so that the field-of-view covers the entire system size, requires the evaluation of $O(N^2)$ interactions for a system of $N$ particles, leading to computational costs at inference and to memory costs during training, which become prohibitive when scaling to large particle numbers. Several solutions have been proposed to address long-range interactions in MLFFs. In classical molecular dynamics (MD), long-range interactions in periodic systems are typically treated using Ewald summation[21]. Inspired by that, Ewald message-passing combines a direct-interaction GNN between particles in real space with a network in the Fourier representation of the periodic particle density[22–27]. Despite the use of Fast Fourier Transforms (FFTs)[24,25], these methods are quite computationally expensive. Another way to enable a global field of view while avoiding oversquashing is to employ global self-attention for each node. Inspired by Large Language Models (LLMs), where their effectiveness is well established[28], this approach updates node representations by aggregating information from the entire graph via a weighted average of the constituent nodes, with the normalized weights calculated from each node-pair[29,30]. The main drawback of global attention is its high computational and memory cost, which scales as $O(N^2)$. By introducing a series of approximations, memory requirements can be significantly reduced[31], enabling linear time scaling[32–34]. In this direction, notable progress has also been achieved in the atomistic domain[7,16,35]. Lastly, the addition of virtual graph elements provides a straightforward way to extend message passing across the entire graph. Although this concept was first introduced in molecular physics almost a decade ago[36], its adoption has been relatively limited[37], despite its demonstrated success in other fields[38–42]. Virtual nodes that aggregate and broadcast information to the entire structure are particularly appealing, as they are characterized by linear time complexity and it has been theorized that they can approximate a self-attention mechanism with some assumptions on the structure of the virtual representation[43]; however, previous implementations were architecture-dependent, using the same message-passing algorithm as the underlying model, and represented the entirety of the system with a single fixed-size vector, limiting the flow of information in the case of arbitrarily large structures[20]. In this work, we present RANGE (Relaying Attention Nodes for Global Encoding): an extension to GNN architectures that can be flexibly combined with a large variety of base frameworks, achieving long-range many-body message-passing for graphs of arbitrary topology. In contrast to existing approaches, RANGE introduces multiple virtual representations with positional encodings that relay information via self-attention, strongly reducing oversmoothing and oversquashing, and scaling linearly with system size. We demonstrate on several examples how RANGE accurately captures long-range behavior and has a small computational cost compared to alternative approaches.

## Results

### The RANGE extension for message-passing neural networks

Building on the standard MPNN paradigm, RANGE introduces a set of virtual nodes as global representations of the underlying graph, to which we refer as master nodes (Fig. 1). Master nodes are virtual nodes (i.e., not associated with the physical position of an atom) connected to all other nodes in the graph that serve as information hubs for message-passing. As we demonstrate later, in practice, each virtual node acts like a "learnable mean field", as it gathers information across different parts of the system and then redistributes it back to the real nodes. After a standard message-passing step, during the aggregation phase, node embeddings are gathered into coarse-grained representations via multi-head self-attention, producing independent representations of aggregated information. This information is communicated back to all graph nodes during the broadcast phase; the base graph nodes can weigh the relative impact of individual master nodes while preserving the relevant information collected during the message-passing step, thanks to the presence of self-loops. Since the master nodes have direct edges to every node of the graph, they capture long-range interactions in a single step, overcoming limitations of strictly local, pairwise, receptive fields, and simultaneously avoiding the oversmoothing that would come with repeating many message-passing steps and the oversquashing that stems from transmitting information through a single finite-dimensional channel, effectively compressing the flow of information. The presence of master nodes dramatically changes the graph's topology towards a small-world structure, in which information can travel long distances with only a few steps[44]. Refer to the Methods section and Supplementary Note 1 for a detailed description of RANGE.

### Capturing long-range behavior beyond the limits of local message-passing

RANGE is an architectural extension that can, in principle, be applied on top of any message-passing framework. We have selected four of the most popular MPNNs for modeling molecular systems[12,17,45,46] as baseline models to perform extensive analyses and demonstrate the performance of RANGE in terms of accuracy and efficiency. Among the architectures we have selected, SchNet[4,5] is the only one that utilizes invariant node representations, while PaiNN[6], So3krates[10], and MACE[8], state-of-the-art in modern molecular modeling, also employ equivariant embeddings, which capture higher-order correlations and lead to higher accuracy in the prediction of both invariant and equivariant properties[47].

To demonstrate the ability of RANGE to capture long-range interactions, we have selected three datasets that were specifically constructed to test the performance of modern MLFFs on systems where these interactions play a major role[48]: a NaCl crystal where a sodium atom is displaced[49], a gold dimer approaching magnesium oxide under periodic boundary conditions[49], and a set of biodimers with different charge and polarity interacting at various distances[50,51]. For each dataset, baseline and RANGE-extended models were trained with 2 interaction layers and a 5 Å cutoff, and tested on the associated task. In the NaCl dataset, a dopant-like effect is created by introducing an additional sodium atom at one edge of the crystal while keeping the charge constant, as shown in Fig. 2a, forcing a redistribution of electronic density. The task is to measure whether a model can correctly predict the resulting energy profile as a sodium atom is displaced from the charged clusters. Similarly, the AuMgO dataset tests the ability of a model to capture long-range interactions by moving a $Au_2$ dimer towards the surface of the periodic MgO crystal, with or without a buried aluminum dopant, as illustrated in Fig. 2b. The dopant shifts the preferred configuration of the dimer from parallel (wetting) to orthogonal (non-wetting) with respect to the surface. The task is to correctly predict the relative stability of the non-wetting configuration with and without the dopant, by reproducing the energy profile as a

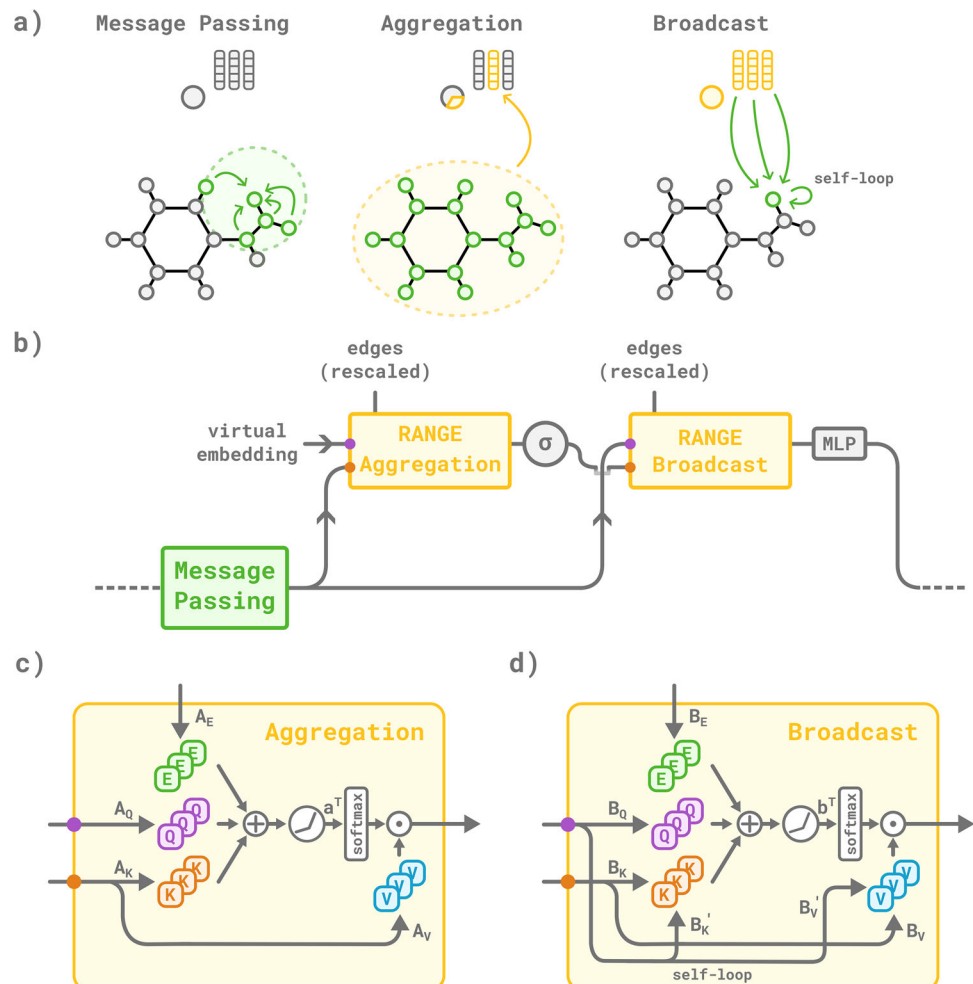

**Fig. 1 | Overview of RANGE.** In **a** the main phases of the RANGE architecture are depicted: a message-passing step propagates information among the nodes of the input graph. The aggregation and broadcast steps collect and redistribute the information. In **b** the updated node representation and the initialized virtual embeddings are fed into the RANGE aggregation block. After an element-wise nonlinearity ($\sigma$), the coarse-grained representation is propagated back via the RANGE broadcast block. Mixing across different heads is performed using a multilayer perceptron (MLP). In **c**, **d**, aggregation and broadcast blocks project senders and receivers onto key and query space, respectively, via the linear layers $A_K$, $A_Q$ and $B_K$, $B_Q$. A positional encoding projected onto the edge space via $A_E$ and $B_E$ is included in the calculation of the attention weights. During the broadcast phase **d**, a memory effect, modeled by self-loops, is introduced for balancing local and global information content inside each graph node. Lastly, the attention weights are used to modulate the contributions of the receivers projected via the $A_V$ and $B_V$ linear layers.

function of the distance between the golden dimer and the surface. With the exception of MACE which correctly predicts the Au-O energy curves, the baseline models fail to predict the energy profile of both the doped and undoped structures; moreover, the energy curves are virtually unaltered by the structural change, indicating a clear inability of the local networks to exchange information at long-range. Remarkably, with the application of the RANGE module, both invariant and equivariant architectures can accurately reproduce the different energy profiles. The dataset of biodimers consists of equilibrium configurations of organic dimers at a 4-15 Å distance. Their interactions span different regimes: at short distances, atoms are kept together by strong electrostatic effects; at longer distances, dispersion forces become dominant. The test data is grouped by charge distribution, and the evaluation of energy and forces is performed for pairs of polar (P), apolar (A), and charged (C) molecules. Notably, only configurations with intermolecular distances extending up to 4 Å beyond the equilibrium distance are used for training, whereas the models are evaluated on dimers with intermolecular separations beyond 4 Å; the task is thus extrapolative. In Fig. 2c, we report the mean absolute errors obtained on energy and forces of the predicted dimers: since standard

message-passing models cannot capture intermolecular interactions beyond their cutoff radius (5 Å), they systematically show larger errors than the RANGE-extended models. RANGE demonstrates robust extrapolation to long-range interaction regimes, with errors up to four times lower than the baseline models when predicting the energies of charged species. Crucially, these tests illustrate that RANGE is not simply a minor adjustment, but it determines a fundamental change in the models' expressive power, by providing access to long-range interactions completely excluded by the local message-passing paradigm.

**Cost-accuracy trade-off of RANGE-augmented models.** Additionally, as a more practical task, we have tested the prediction of intramolecular long-range interactions by training atomistic MLFFs on the Aquamarine (AQM) dataset[52], which contains a variety of challenging structures, with sizes ranging from 30 to 92 atoms. In Supplementary Table 4, we report additional results on QM7-X[53], a dataset consisting of smaller structures of up to 23 atoms. As further discussed in Supplementary Note 2, all reference data explicitly include the quantum treatment of long-range effects via many-body dispersion[54,55]. In

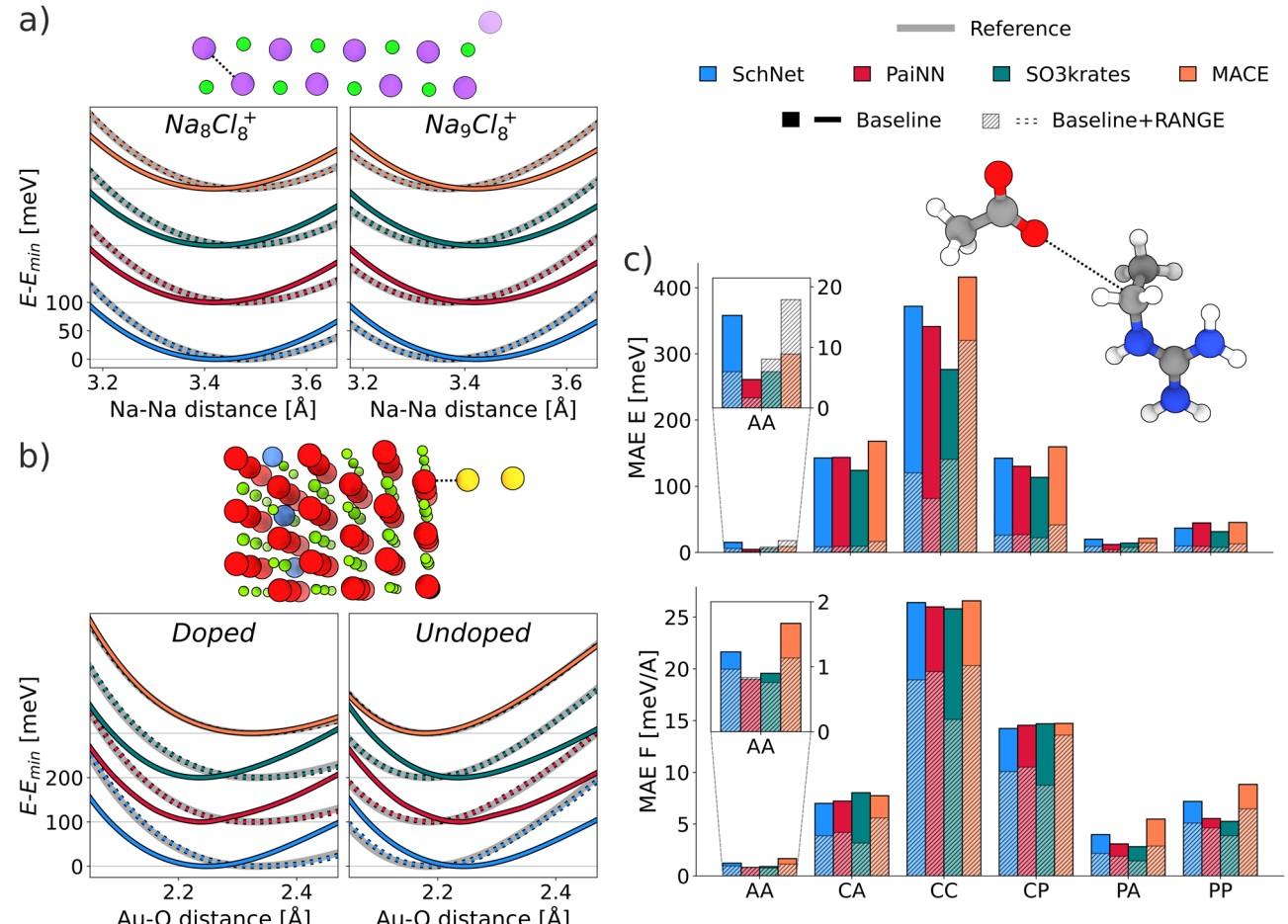

**Fig. 2 | Long-range tasks.** RANGE-extended message-passing neural network (SchNet[4,5], PaiNN[6], SO3krates[10], and MACE[8]) are compared to their baseline version in a set of tasks designed to assess the ability to model long-range effects. **a** Relative energies of $Na_8Cl_8^+$ and $Na_9Cl_8^+$ systems (Na in purple, Cl in green) as a function of Na-Na distance, as highlighted in the structure figure. As the furthest Na atom is added or removed, the charge is kept constant. Offsets are added to distinguish the different models. **b** Relative energy of periodic MgO as the $Au_2$ dimer approaches the surface of the Al-doped and undoped crystal (Mg in red, O in green, Al in blue, and Au in yellow) as a function of the Au-O distance, as highlighted in the structure figure. Offsets are added to distinguish the different models. Structures and reference energy in **a**) and **b**) are taken from[48]. **c** Mean absolute error (MAE) of energy and forces of different organic dimers in an extrapolation task beyond the distance cutoff explored during training. Results are divided by the electronic distribution of the molecules in the dimers: (A)polar, (P)olar, (C)harged. The depicted molecule is extracted from the CC test set. The reported legend is shared among all panels. Source data are provided as a Source Data file.

Fig. 3a, b, we compare the mean absolute error (MAE) on the forces against inference time and peak memory usage at inference for the AQM dataset using all MPNN architectures and their RANGE counterparts. RANGE consistently outperforms all baseline models at any given cutoff, while increasing the cutoff only marginally improves the performance of the baseline models. The errors of the baseline models quickly saturate, indicating the presence of information bottlenecks, i.e., oversquashing. However, even the RANGE models with the shortest cutoff outperform the baselines with the longest reach. The time and memory requirements of the RANGE implementation scale linearly with the system size: at any given cutoff, RANGE increases the baseline model inference time and memory peak by a constant and relatively small amount. The numerical values are provided in Supplementary Table 6. This behavior aligns with the theoretical complexity of the method. The expected linear scaling is validated by measuring the inference time as a function of the number of atoms in test systems up to 70 000 atoms, as reported in Supplementary Fig. 2, and during molecular dynamics simulations, as reported in Supplementary Fig. 4. Although it was recently suggested that adding global aggregations could only lead to better performances[37], we observe that, if these are left unconstrained, the attention weights of multiple master nodes can become degenerate, leading to a degradation of accuracy with an increasing number of master nodes (as shown in Supplementary Fig. 3 and Supplementary Tables 4 and 5). To address this issue, we introduce a regularization term to dynamically allocate the number of master nodes as a function of the system size, effectively acting as an expandable space for storing global information (Supplementary Note 1). We stress that, for both datasets in Supplementary Fig. 3, all regularized RANGE models lie below the smallest possible MAE that is achievable by naively increasing the message-passing cutoff, even in QM7-X, where the large majority of compounds is fully included within the largest cutoff value tested (7 Å).

**Comparison with alternative long-range approaches.** We have also tested our approach against existing long-range corrections for atomistic MPNNs. As a notable example among the Ewald-based methods, Ewald MP[22] projects the node embeddings onto the reciprocal space via Fourier expansion and applies a learned frequency filter to specifically select long-range interactions; after transforming the embeddings back to the real space, the additional contribution is added to the prediction of the baseline model. The more recent Neural P³M[25] improves upon Ewald MP and employs the computationally efficient Fast Fourier Transform (FFT) in place of explicit Ewald summation. Neural P³M demonstrated a systematic improvement in terms of

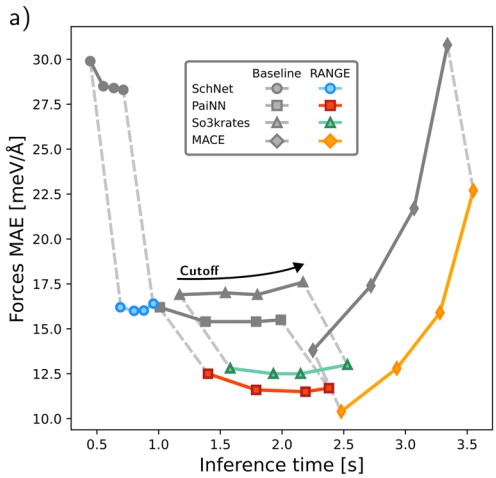
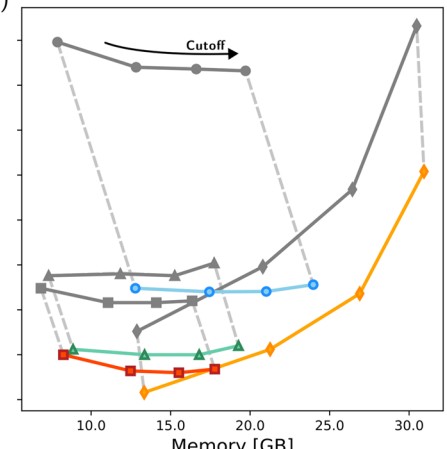

**Fig. 3 | Cost-accuracy trade-off.** Baseline and RANGE-augmented models (SchNet[4,5], PaiNN[6], SO3krates[10], and MACE[8]) are evaluated on the AQM dataset and reported for different message-passing cutoffs (5, 7, 9, 12 Å) and 3 interaction blocks (2 for MACE). The mean absolute error (MAE) of the predicted forces on the test set is reported against a) inference time and b) peak memory usage at inference. As stated in Supplementary Note 3.2, batch sizes of 512 for SchNet and 128 for PaiNN, SO3krates, and MACE were used. Source data are provided as a Source Data file.

accuracy compared with Ewald MP[25]. RANGE, Ewald MP, and Neural P³M can be applied to virtually any MPNN out-of-the-box; therefore, we compare their performance by training the PaiNN MPNN with a cutoff of 5 Å on different portions of the MD22 dataset[56], corresponding to ~ 70 000 simulation frames of docosahexaenoic acid (DHA), ~ 6 000 frames of the buckyball catcher (CC), and ~ 5 000 frames of the double-walled nanotube (DW). At 56, 148, and 370 atoms, respectively, these systems not only represent a considerable challenge in terms of accuracy for the role that long-range interactions play in their stability, but also in terms of computational efficiency. Table 1 shows that both RANGE and Neural P³M achieve significantly lower MAEs with respect to the baseline models. On the other hand, the improvement over the baseline models obtained with Ewald MP is smaller. We note that, while in terms of accuracy, RANGE and Neural P³M result practically equivalent, RANGE outperforms Neural P³M both in computing time and maximum memory allocated. The fact that Neural P³M becomes competitive for the large double-walled nanotube suggests that a large prefactor contributes to the calculation of the FFT grid in Neural P³M at inference time. However, in the large system regime when Neural P³M becomes competitive, the memory cost of building such a large grid makes it unfeasible to use for molecular dynamics (MD) simulations because, as molecular geometries evolve, the grid needs to be calculated ex-novo at each step.

## Molecular Dynamics Simulations with RANGE

An important requirement for atomistic force fields is the continuity of energies and forces with respect to the positions of the input coordinates. This property is well-known and is often obtained in standard MLFF models through the introduction of continuous filtering convolutions[4], which leverage smooth cutoffs to rescale the messages. Since our main objective is to aggregate and broadcast information between a set of master nodes and the entire underlying graph, this approach is not applicable at the master node level, as the graph boundaries are not well defined: any kind of direct distance-based encoding would inherently lead to the introduction of a limited field of view given by the pairwise distribution of the training dataset. This would result in limited transferability of the method for systems with large node delocalization. In RANGE, we address this issue by introducing a continuous SE(3)-invariant positional encoding, where arbitrarily large distances are continuously mapped to the [0, 1] interval and projected into a high-dimensional space via an expansion into Gaussian radial basis functions[4]. To assess the stability of our

approach, we performed 16 ns-long MD simulations of the DHA, buckyball catcher, and double-walled nanotube systems in the gas phase using the PaiNN models trained with RANGE. For the three systems, 10, 5, and 2 independent runs were performed using a Langevin integrator and a 1 fs timestep. None of the trajectories exhibited any sign of instability during the simulations. Figure 4 presents a representative 2 ns segment of the MD trajectories for each system, illustrating the ability of our architecture to generate stable and physically meaningful trajectories that effectively explore complex potential energy landscapes.

## Interpretability of RANGE

The magnitude of self-attention weights is often used to interpret deep learning models and understand which features are the most relevant for the models' outputs[57-60]. The additive attention mechanism used in RANGE (Supplementary Note 1) can provide increased flexibility with respect to the more popularized dot-product attention[30]; additionally, it has been suggested that this form of attention also leads to more interpretable neural networks[61]. Since our model preserves independence of the attention heads during each aggregation-broadcast cycle (see Methods section and Supplementary Note 1), we can explore the relative importance of individual atoms within a given step. As shown in Supplementary Fig. 5, performing a singular value decomposition (SVD) analysis on the attention weight distribution of the DHA, CC, and DW models discussed in the previous section reveals that each of the 8 attention heads exhibits a distinct, dominant degree of freedom, or clustering strategy. Figure 5 shows the principal component of the SVD analysis (i.e. the singular vector corresponding to the largest singular value) of the attention weights during the aggregation and the broadcast steps, mapped on the graph nodes. In the figure, we report results for a single attention head to illustrate the flow of information across the three systems. These figures show how the information clustered during the aggregation step is redistributed to the graph nodes during broadcasting, highlighting the non-local nature of the clustering procedure and the inherent N-body nature of the node-node communication via virtual embeddings. This is akin to a mean-field effect, where the aggregation step outputs a weighted average of the components, and the nodes feel an effective interaction via the master node during broadcast. In this setting, each attention head produces a different learnable aggregated representation that does not rely on predefined heuristics as typically required by clustering strategies and allows for context-dependent weighting of information.

**Table 1 | Comparison between RANGE, Ewald MP, and Neural P³M**

| | Model | MAE energy [kcal/mol] | MAE forces [kcal/mol/Å] | Rel. inference time [a.u.] | Rel. memory peak [a.u.] |
|---|---|---|---|---|---|
| DHA | Baseline | 0.199 ± 0.004 | 0.238 ± 0.004 | - | - |
| | Ewald MP | 0.191 ± 0.003 | 0.243 ± 0.002 | 3.351 ± 0.021 | 1.695 ± 0.004 |
| | Neural P³M | **0.113** ± 0.007 | **0.126** ± 0.007 | 2.802 ± 0.118 | 3.967 ± 0.010 |
| | RANGE | **0.110** ± 0.002 | 0.160 ± 0.004 | **1.471** ± 0.007 | **1.216** ± 0.003 |
| CC | Baseline | 0.37 ± 0.06 | 0.285 ± 0.006 | - | - |
| | Ewald MP | 0.36 ± 0.08 | 0.289 ± 0.006 | 1.49 ± 0.02 | 1.544 ± 0.001 |
| | Neural P³M | **0.18** ± 0.06 | **0.182** ± 0.034 | 1.40 ± 0.05 | 1.789 ± 0.001 |
| | RANGE | **0.22** ± 0.03 | **0.211** ± 0.005 | **1.24** ± 0.02 | **1.166** ± 0.001 |
| DW | Baseline | 0.96 ± 0.19 | 0.71 ± 0.02 | - | - |
| | Ewald MP | 0.89 ± 0.11 | 0.73 ± 0.03 | **1.44** ± 0.02 | 1.367 ± 0.003 |
| | Neural P³M | **0.40** ± 0.08 | **0.56** ± 0.03 | 1.59 ± 0.14 | 2.415 ± 0.003 |
| | RANGE | **0.47** ± 0.06 | **0.52** ± 0.02 | 1.57 ± 0.03 | **1.125** ± 0.002 |

Mean absolute error (MAE) of energies and forces, relative inference times, and peak memory usage at inference on large molecular systems from the MD22 dataset[56]: docosahexaenoic acid (DHA), buckyball catcher (CC), and double-walled nanotube (DW). The reported values are averages of four independently trained (with different dataset seeds) PaiNN[6] baseline models and their extensions with RANGE, Ewald MP[22], and Neural P³M[25]. The best architectures for each system, within the margin of error, are marked with bold lettering. Source data are provided as a Source Data file.

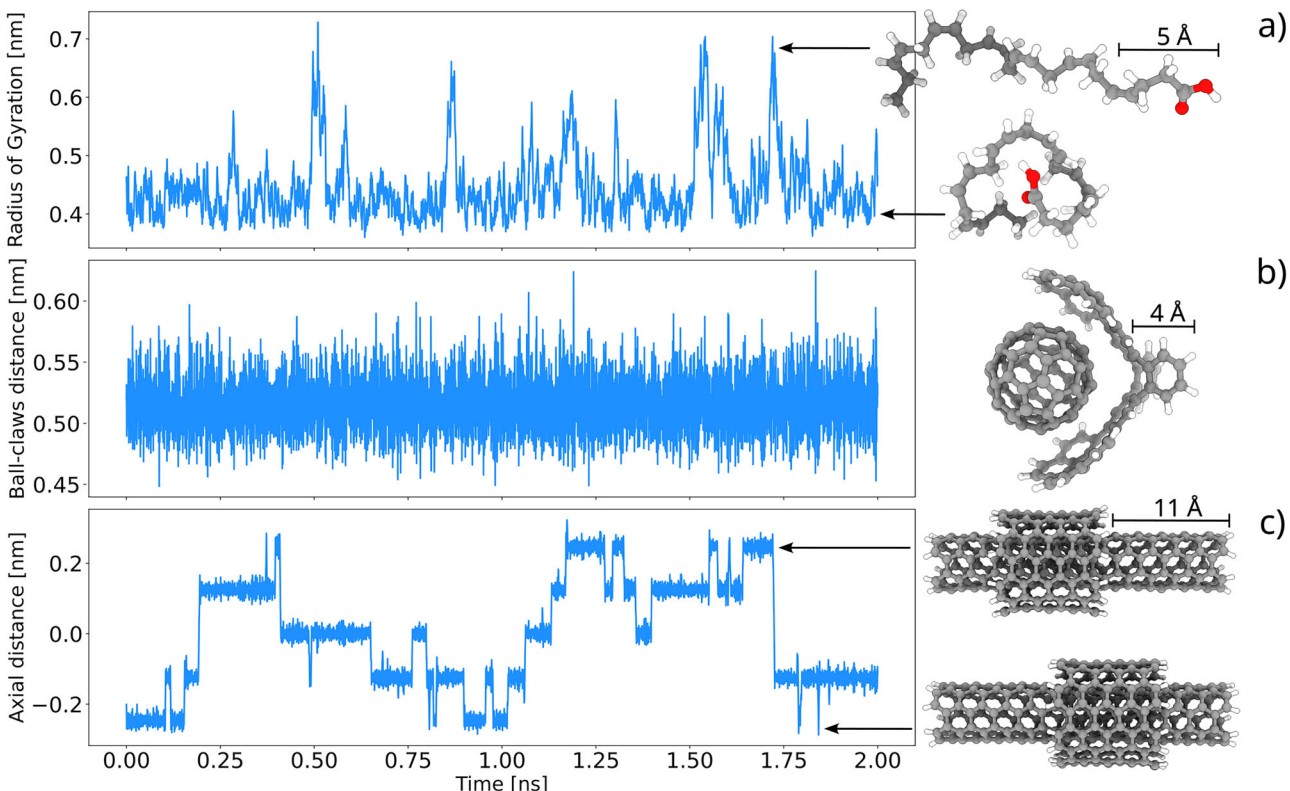

**Fig. 4 | Molecular dynamics simulation timeseries obtained with the RANGE architecture.** All simulations were performed for 16 ns with the RANGE architecture applied on PaiNN[6] with a 5 Å cutoff trained on docosahexaenoic acid (DHA), buckyball catcher (CC), and double-walled nanotube (DW) from the MD22 dataset[56]. **a** Radius of gyration timeseries for DHA, showing that the simulation explores both extended and compact molecular conformations; **b** ball-claw distance of CC, measured between their centers of mass; **c** distance of the external cylinder of DW relative to the axial center of the inner cylinder. Five distinct metastable states are explored. Representative structures corresponding to different metastable regions are shown. Source data are provided as a Source Data file.

## Discussion

In this work, we propose RANGE: an architectural extension that can be combined with any GNN to recover long-range N-body interactions. This is achieved in a two-stage process: global aggregation of the information into virtual embeddings, followed by broadcasting the coarse-grained representations onto the nodes of the original graph. With respect to other approaches that employ virtual aggregations, we make use of multiple, dynamically activated virtual nodes to extend the capacity of the embeddings and scale up to larger systems, and a self-attention mechanism that has been shown to reduce over-squashing in GNNs. We have demonstrated our framework by combining it with state-of-the-art invariant and equivariant GNN architectures, namely SchNet, PaiNN, MACE, and So3krates. In a set of tasks specifically designed to test MPNNs' ability to explicitly model

## Principal component of attention weights

**Fig. 5 | Principal component of attention weights.** Principal component of the singular value decomposition on the attention weight distribution during aggregation and broadcast is reported for a selected attention head of the RANGE architecture applied to PaiNN[6] and trained on **a** docosahexaenoic acid, **b** buckyball catcher, and **c** double-walled nanotube from the MD22 dataset[56]. Source data are provided as a Source Data file.

long-range interactions, all baseline models fail to recover the correct behavior. RANGE-based architectures, on the other hand, are not only able to correctly reproduce behavior they are presented during training, but are also able to extrapolate long-range effects outside of their training data. Additional tests on real-world systems show that RANGE outperforms the baseline models in terms of accuracy and, with its linear time complexity, outcompetes other popular solutions for the inclusion of nonlocal effects, such as Ewald-based networks, in terms of scaling and memory usage. The edge features in our proposed model are designed in a way that guarantees the transferability across different sizes and preserves the continuity of the energy with respect to the atomic positions, a required property in an MLFF. We report simulation trajectories for three large systems in the MD22 dataset that remain stable for more than 15 ns, during which the model effectively samples different metastable configurations. An SVD analysis of the attention weights of the virtual embeddings during the aggregation and broadcast phases reveals the presence of a single degree of freedom for each attention head, suggesting a well-defined clustering strategy; moreover, the simultaneous activation of multiple nodes spanning the entire system confirms that the distributed information is inherently N-body, leading the graph nodes to produce an adaptive mean-field effect, clustering different parts of the system during the two-phase process. The results presented show that oversquashing significantly reduces the reach of ordinary MPNNs, leading to saturation of the MAE at large cutoff values and rendering them incapable of modeling long-range effects in molecular systems. Equivariant architectures still suffer from this phenomenon, suggesting that the gains offered by including equivariant information are inherently short-range. Our results demonstrate the potential of attention-based virtual aggregations to improve the overall description via MPNNs of delocalized, many-body molecular systems by creating long-range

communication channels. In particular, RANGE-like implementations that dynamically expand the capacity of the virtual embeddings via a learned regularization parameter are able to efficiently scale up the accuracy gains to very large systems. This is achieved with a small computational overhead, constant with respect to the cutoff, and linear scaling with system size. Future work will focus on investigating the applicability of RANGE to complex environments, such as solvated biomolecules and materials, where long-range interactions play a crucial role.

## Methods
### The RANGE architecture
Consider a graph $\mathcal{G}$, defined by a set of $N$ nodes $\mathcal{V}$ and a set of edges $\mathcal{E} = \{\mathbf{e}_{ij}\}_{i,j=1}^{N}$, with $\mathbf{e}_{ij} \in \mathbb{R}^f$. In a standard MPNN, a learnable feature or embedding $\mathbf{h}_i^{(0)} \in \mathbb{R}^h$ is defined for every node, and sequentially updated at each interaction layer $t$ via

$$\mathbf{h}_i^{(t+1)} = \upsilon_t(\mathbf{h}_i^{(t)}, \mathbf{m}_i^{(t)}), \tag{1}$$

where $\upsilon_t$ is a differentiable update function; $\mathbf{m}_i^{(t)}$ is the aggregation of messages to the $i$-th node from its neighbors, defined as

$$\mathbf{m}_i^{(t)} = \bigoplus_{j \in \mathcal{N}(i)} \mu_t(\mathbf{h}_i^{(t)}, \mathbf{h}_j^{(t)}, \mathbf{e}_{ij}), \tag{2}$$

where $\mu_t$ is a differentiable function and $\bigoplus_{j \in \mathcal{N}(i)}$ is a pooling operation over the neighbors $\mathcal{N}(i)$ of node $i$ designed to respect the graph symmetries. After $T$ interaction layers, a learnable readout function $\mathcal{R}(\{\mathbf{h}_i^{(t)}\}_{t=0}^{T})$ is used to make predictions on the target values. We define a master node $M$ of $\mathcal{G}$ as a virtual node that is connected with all elements in $\mathcal{V}$ via the set of edges $\mathcal{E}(M) = \{\mathbf{E}_i \mid \mathbf{E}_i \in \mathbb{R}^f\}_{i=1}^{N}$, with the purpose of taking long-range interactions into account by aggregating all the nodes in the graph and redistributing information. To allow for a consistent definition of the edges connecting all the graph nodes to a master node, both reside within the same space; for our application on metric graphs such as those used in MLFFs, we position each master node at the geometric center of the graph. Message-passing through $M$ consists of an aggregation and broadcast phase, as illustrated in Fig. 1. The former aims at harvesting information from each node embedding, collecting it in a compressed space via a GATv2-inspired multi-head self-attention mechanism[29,30,62]; the latter redistributes the coarse-grained information to each node of the graph via a self-attention mechanism that parses all the aggregated representations. Together, aggregation and broadcast enable dynamical long-range communication between nodes. Further details on the architecture are provided in Supplementary Note 1.

### Data selection and preparation
The datasets used in this work are publicly available and consist of energies and forces calculated at the DFT level of theory. Additional information on the preparation and composition of the datasets can be found in Supplementary Note 2.

### Training and MD simulations
The models were trained and simulated using the MLCG package[12]. The models were trained for 200 epochs for all datasets except MD22, where they were trained for up to 500 epochs. All MACE-based models were trained for up to 600 epochs. A combined loss of energies and forces with the AdamW optimizer[63] was used. Simulations were performed using a Langevin integrator at 300 K with a fs timestep. Additional details are available in Supplementary Notes 3 and 4.

## Data availability

The split files for the generation of the datasets, the configuration files for training and simulation, and all the MD trajectories are publicly available at https://zenodo.org/records/18390311[64]. Source data are provided with this paper.

## Code availability

The RANGE codebase is publicly available at https://github.com/ClementiGroup/range[65].

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

## Acknowledgements

We thank members of the Clementi's group for insightful discussions and comments on the manuscript. We gratefully acknowledge funding from the Deutsche Forschungsgemeinschaft DFG (SFB/TRR 186, project A12; SFB 1114, projects B03, B08, and A04), MATH plus (projects AA1-6 and AA2-20), the National Science Foundation (PHY-2019745), the Einstein Foundation Berlin (project 0420815101), the Bundesministerium für Bildung und Forschung BMBF (project FAIME 01IS24076), and computing time provided on the supercomputer Lise at NHR@ZIB as part of the NHR infrastructure (project beb00040). The authors also gratefully acknowledge the Gauss Centre for Supercomputing e.V. (www.gauss-centre.eu) for funding this project by providing computing time on the GCS Supercomputer JUWELS at Jülich Supercomputing Centre (JSC) (project mlcg). We thank the HPC Services of FUB-IT and the Physics department of Freie Universität Berlin; we are especially grateful to Jens Dreger for helping us with the computational setting.

## Author contributions

A.C., J.V., F.N., and C.C. designed the RANGE architecture. A.C. and J.V wrote the code. A.C., J.V., L.G., E.R., and Z.E.-M. ran the experiments. A.C., J.V., L.G., E.R., Z.E.-M., F.N., and C.C. designed figures, experiments, and contributed to writing the manuscript.

## Funding

## Competing interests

The authors declare no competing interests.
