## [Transparent Peer Review file · Nature Communications]

Extending the Range of Graph Neural Networks with Global Encodings

Corresponding Author: Professor Cecilia Clementi

Version 0:

Reviewer comments:

Reviewer #2

(Remarks to the Author)

I had reviewed this paper already for its [editorial note: journal name redacted] submission, expressing a very favorable opinion. I must say that this version is significantly improved. The authors did a great job in giving a good intuition why the introduction of master virtual nodes helps into modelling long-range interactions. The additional validation and benchmarks are also impressive, both in the amount and in the quality of the results (see for instance Fig. 2). The authors demonstrated in a convincing way that their proposed method is generalizable by showcasing its application on various systems ranging from materials to biomolecules.

This manuscript is of the highest quality and contains a very significant contribution to the computational molecular community.

I recommend the publication of this manuscript as it is without any further delay.

(Remarks on code availability)

Reviewer #3

(Remarks to the Author)

The authors have added new tests and new baselines, which makes the revised manuscript more substantial. I am not convinced that these tests are the best ones, but there are a number of papers and preprints now coming out from the community on adding long range interaction to MLFFs and this work certainly deserves to be seen, so I won't stand in its way.

(Remarks on code availability)
